# Seroprevalence for Measles, Varicella, Mumps and Rubella in the Trainee Obstetric Population: A Survey in Southern Italy

**DOI:** 10.3390/vaccines12030335

**Published:** 2024-03-20

**Authors:** Brunella Perfetto, Giovanna Paduano, Elena Grimaldi, Vincenza Sansone, Giovanna Donnarumma, Gabriella Di Giuseppe

**Affiliations:** 1Department of Experimental Medicine, Section of Microbiology and Clinical Microbiology, University of Campania “Luigi Vanvitelli”, 80138 Naples, Italy; brunella.perfetto@unicampania.it (B.P.); giovanna.donnarumma@unicampania.it (G.D.); 2Department of Experimental Medicine, Section of Hygiene, Occupational Medicine and Forensic Medicine, University of Campania “Luigi Vanvitelli”, 80138 Naples, Italy; giovanna.paduano@studenti.unicampania.it (G.P.); vincenza.sansone@unicampania.it (V.S.); 3AUO Virology and Microbiology, Azienda Universitaria Policlinico “Luigi Vanvitelli”, 80138 Naples, Italy; elena.grimaldi@unicampania.it

**Keywords:** seroprevalence, measles, varicella, mumps, rubella, trainee obstetrics, survey, Italy

## Abstract

Background: Despite the efforts of the World Health Organization, some childhood viral diseases, for which there is already an effective vaccine, have not yet been eradicated. Among these, we find varicella, mumps, measles, and rubella, which although in most cases have a benign course, can in some cases be responsible for infectious outbreaks, especially in nosocomial settings. The aim of this study was to verify the immunological situation of a cohort of trainee obstetricians in Campania regarding varicella, mumps, measles, and rubella to be used as an example for the evaluation of possible preventive strategies to avoid infectious outbreaks. Methods: All the samples collected and sent to the laboratory were eligible for analysis and have been included in the study. Specific IgG for varicella, measles, mumps, and rubella were assayed on serum samples taken from 517 trainee obstetricians using the enzyme-linked immunosorbent assay (ELISA) technique. The seropositivity results were statistically analyzed by correlating them to age group and gender. Results: The results obtained show that a percentage of trainee obstetricians tested do not have an effective immunological coverage against at least one of the vaccine-preventable diseases considered, especially for mumps. Conclusions: Therefore, it is proposed to extend surveillance to other professionals in contact with frail patients and increase awareness of vaccination campaigns.

## 1. Introduction

Measles, mumps, rubella, and varicella are viral diseases often linked to childhood. Although they are highly contagious, there is a safe and effective vaccine against them. They are still not eradicated, despite the efforts of World Health Organization (WHO) programs designed to eliminate these vaccine-preventable pathologies worldwide. Often, they re-emerge causing recurrent outbreaks in both developed and industrialized countries. Furthermore, the Centers for Disease Control and Prevention (CDC) suggests vaccinating females at least one month before pregnancy for all these pathologies to avoid risks for fetuses and newborns [1]. In addition, in 2020 and 2021 due to the COVID-19 pandemic, many countries reported a decline in vaccination coverage. This, together with the lifting of measures to contain COVID-19 such as physical distancing, hand hygiene, and mask-wearing, has been responsible for an increased transmission, for example, of measles [2,3].

Measles is a Paramyxovirus causing fever and coughs. During the progression of the pathology, ulcerative white lesions on the buccal mucosa known as Koplik’s spots appear; these are pathognomonic of the disease. A few days after the beginning of the symptomatic phase, a morbilliform skin rash appears. Often, due to an induced transient immunosuppression, the patients develop bacterial superinfections; these can be responsible for pneumonia, otitis, bronchitis, or diarrhea. More complications can appear at the expense of the nervous system, such as encephalitis and sub-acute sclerosing panencephalitis (SSPE) [4]. In 2018–2019, there was a resurgence of measles cases, which resulted in a fair number of deaths, subverting the efforts made by the regional health system aimed at eliminating the spread of the virus [5]. Unfortunately, in Europe, the measles virus continues to circulate. This is due, according to the epidemiological reports, to the failure to achieve considerable vaccination coverage. To avoid cases of resurgence and the free spread of the virus, a vaccination coverage of 95% for two doses of vaccine would be necessary [3,6]. A Paramyxovirus is also responsible for mumps. The disease is characterized by different symptoms, such as fever and parotid inflammation responsible for puffy, tender, and swollen cheeks. The virus has a marked tropism for the nervous system, with half of cases leading to aseptic meningitis or encephalitis. Other complications may occur, such as pancreatitis, orchitis, and deafness [7]. Rubella is characterized by a cutaneous rash accompanied by fever. The rubella virus normally causes a self-resolving disease with a benign outcome, but its severity increases if it affects pregnant women, especially during the first three months of pregnancy. The virus, in fact, is able to cross the placental barrier, causing serious damage and death to the fetus. Congenital rubella syndrome (CRS) includes all the congenital defects that the virus causes to the fetus: eye anomalies such as cataracts, retinopathy, microphthalmia, pigmentary, and congenital glaucoma; sensorineural hearing loss; heart defects such as ventricular septal defects and pulmonary stenosis; other lifelong disabilities such as autism, encephalitis, and thyroiditis. The estimated global annual number of CRS cases declined gradually during 1996–2016 from 121,000 to 32,000 [8,9].

Frau et al. [2] reported some suggestive data on the incidence of these pathologies in the period preceding the one in which the vaccine was made mandatory. Concerning varicella, the last Italian report on epidemiological trends, published in 2013, reported 59,388 cases. In 2017, more than 4400 measles cases were reported in Italy in infants below one year of age and in 7% of healthcare workers (HCW); among these, in three cases the patients died and two had encephalitis as a complication. Again in 2017, 65 rubella cases were reported, double those reported in the previous year, and 829 cases of mumps were reported by the European Centre for Disease Prevention and Control (ECDC) [10].

The MMR vaccine for measles, mumps, and rubella was included in Italy from 1987 and obligatory from 2017. It is considered highly effective after two doses, the first around the 12th and 15th month of age and the second around 4th and 6th years of age. Currently, Merck is the sole manufacturer of FDA-approved formulations of the MMR vaccine, which are presented in two formulations, a monovalent e.g., Varilrix (GlaxoSmithKline Biologicals S.A- Rixensart, Belgio), M-M-R^®^II and Varivax (Merck & Co., Inc., Rahway, NJ, USA) or quadrivalent form (ProQuad^®^; Priorix-Tetra; GSK, Belgium, or ProQuad; Merck & Co., Inc., USA) which also contains a fourth vaccine in its formulation for varicella [11].

Varicella-zoster virus (VZV) is responsible for both varicella, also known as chickenpox, and herpes zoster (HZ), also known as shingles. Varicella virus has a marked neurotropism, so can lie dormant in the nerve ganglia of patients that have been affected by varicella and subsequently can cause the onset of shingles. Normally, varicella gives a mild disease, more severe in adults, that commonly confers lifetime immunity. Its symptoms are a cutaneous vesicular rash, generally limited to the neck and trunk, and fever. A few severe complications may occur in immunocompromised patients, such as pneumonia, osteomyelitis, and neurological problems. In pregnant women it may be transmitted to the fetus and responsible for spontaneous abortion, preterm birth, and a congenital syndrome characterized by several anomalies (CVS) [12]. As reported above, a vaccine is available for this virus and has been licensed for use since 1984. The first commercially available formulation was a frozen formulation, subsequently updated to a refrigerator-stable varicella vaccine in 1991, and licensed for use in 1994. Although not universally adopted, WHO recommends it [6,13].

Healthcare workers, due to their job, have a high risk of contracting infections compared to normal people. This makes them an important study group for the prevention of infectious outbreaks at hospital level [14,15,16,17]. In Italy, it has been described that coverage for recommended vaccinations among HCWs is very low [18], therefore the aim of the present research was to evaluate the immunization status for vaccine-preventable disease such as measles, mumps, rubella, and varicella in a cohort of 517 trainee obstetricians attending at the University of Campania “Luigi Vanvitelli”, considering that they are the future healthcare workers who will have close contact with pregnant women and unborn children. Investigating the presence of serological antibodies could be useful in order to develop preventive strategies to avoid nosocomial outbreaks, potentially including, by the occupational medicine service, a vaccination plan for those HCWs who lack or do not show evident immunization.

## 2. Materials and Methods

### 2.1. Participants and Study Design

The study participants were trainee obstetricians on of base and post-base courses courses recruited between 2017 and 2022 at the Health surveillance service of the University of Campania Luigi Vanvitelli, located in Naples, Southern Italy. A random sample of 517 trainee obstetricians was recruited and underwent venous blood sampling. After collection of a consent form, the blood samples were collected and centrifuged within two hours of collection, aliquoted, and stored at 20 °C until tested. All the samples collected and sent to the laboratory were eligible for analysis and have been included in the study.

### 2.2. Laboratory Methods

The levels of specific IgG for varicella, measles, mumps, and rubella were evaluated on serum samples. For a quantitative determination of the specific antibodies, the enzyme-linked immunosorbent assay (ELISA) technique was used. For varicella, measles, and mumps IgG determination, the analysis was conducted using the ELISA Kits from DRG Instrument GmbH, Dortmund, Germany; while for rubella IgG determination, has been used the ELISA Kit from DIESSE Senese Diagnostic, Siena, Italy.

Briefly, microtiter plates were coated with specific highly purified antigens of the viruses; these are able to link, in the serum samples tested, to the corresponding antibodies. To remove all unbound samples, the microtiter plates were washed three times and then an anti-human IgG conjugated with horseradish peroxidase (HRP) was added to the wells. This bound the captured antibodies of the samples. The unbound conjugate was then removed with three more washes. The addition of the substrate, Tetramethylbenzadine (TMB), made the immunocomplex formed by binding the conjugate visible, producing a colorimetric blue reaction. The next addition of sulfuric acid stopped the reaction and produced a yellow endpoint colour. Thanks to a plate spectrophotometer, the optical density of each control, cutoff, and sample was read at 450/620 nm. Internal anti-rubella, varicella, measles, and mumps virus IgG quality control samples were tested at least twice in each plate. The intensity of the colour obtained in each well is proportional to the level of specific antibodies in the samples.

Overall, the diagnostic probability of a negative score in the absence of the specific analyte (specificity) is in a confidence interval between 85.75% and 99.49%. The diagnostic probability of a positive score in the presence of the specific analyte (sensitivity) is in a confidence interval between 89.4% and 96.43%.

### 2.3. Data Analysis

The data were stratified by gender and by age. The results were calculated in arbitrary units (AU) due to the ratio between the sample (mean) OD value/the cutoff OD value. Distribution of anti-IgG titers was assessed as one of three categories: Seronegative, Equivocal susceptibility and Seropositive.

For IgG anti rubella titers, the values considered were: negative < 0.7 AU, equivocal 0.7 to <1.3 AU, and positive > 1.3 AU. For IgG anti varicella, measles, and mumps titers the values considered were: negative < 9 AU, equivocal 9 to <11 AU, and positive >11 AU. The equivocal results have been considered negative and included in the statistical analysis as susceptible.

### 2.4. Statistical Analyses

A descriptive (proportions, means, standard deviations) and inferential (bivariate analysis) statistic description were used to analyze data using Stata software version 17 [19]. First, the main characteristics of participants were summarized by conducting a descriptive statistics analysis (Table 1). Second, a univariate analysis was performed through the chi-square test and *t*-test to evaluate the association between seroprevalence and age and gender (Table 2). All reported *p* values equal to or less than 0.25 were considered of statistical significance.

## 3. Results

Of the 517 trainee obstetricians included in the study, the results showed that 87% were female and the average age was 26.3 years (range 19–53). The protective IgG values were found in 91.7%, 83.9%, 82%, and 76.2%, respectively, for measles, varicella, rubella, and mumps (Table 1).

Our population shows a lower antibody coverage for mumps, compared to the other tested viruses. Furthermore, this seronegative is confirmed without difference in the two genders. The immune status of the study population, according to age and gender, is shown in Table 2.

Analyzing measles, there were differences described for the participants’ gender and age. Regarding gender, the seroprevalence was the lowest in females (91.6%) and the highest in males (92.5%). Nevertheless, this difference was not statistically significant (*p* = 0.543). The seroprevalence was the lowest in trainee obstetricians aged from 30 to 36 years (89.5%), and the highest in those aged ≥37 years (95.5%), and this difference was statistically significant (*p* = 0.208) (Table 2). Analyzing age as a continuous variable and comparing the mean, there was not a statistically significant difference (*p* = 0.968).

Analyzing varicella and rubella, there were significant differences described when the participants were assessed according to age groups. Indeed, the seroprevalence was the lowest in trainee obstetricians aged ≤23, 77% and 78.5% for varicella and rubella, respectively, and the highest in those aged from 30 to 36 years, 93.9% and 90.4% for varicella and rubella, respectively. Moreover, these differences were statistically significant, with *p* = 0.011 for varicella and *p* = 0.006 for rubella. Analyzing age as a continuous variable and comparing the mean, there was a statistically significant difference (*p* = 0.005 for varicella; *p* = 0.004 for rubella). Instead, assessing participants according to gender, there were no significant differences described for seroprevalence in varicella and rubella (Table 2).

Analyzing mumps, there were significant differences described for the participants’ gender and age. Regarding gender, the seroprevalence was the lowest in females (75.8%) and the highest in males (79.1%). Nevertheless, this difference was not statistically significant (*p* = 0.427). The seroprevalence was the lowest in trainee obstetricians aged ≤23 (70.8%), and the highest in those aged from 30 to 36 years (85.1%), and this difference was statistically significant (*p* = 0.089), confirmed analyzing age as a continuous variable and comparing the mean (*p* = 0.031) (Table 2).

## 4. Discussion

Italy is currently enforcing the National Plan of Vaccine Prevention 2023–2025 (PNPV), approved at the State-Regions Conference on 2 August 2023. The PNPV is intended to act as a blueprint, recognizing as a public health priority the reduction or elimination of the burden of vaccine-preventable infectious diseases through the identification of effective and standardized strategies to be implemented throughout the country. Among the main objectives of the PNPV are the following: to improve surveillance of vaccine-preventable infectious diseases, to fortify communication in the field of vaccines, to promote a culture of vaccinations and training in vaccinology in HCWs [20]. In Italy, vaccinations against rubella, measles, and mumps have been indicated as mandatory since 2017 with Decree Law n. 73 requiring urgent provision in the area of vaccine prevention [21].

Monitoring of seroprevalence levels, especially among HCWs, is an important tool to prevent the spread of infections, and the present results have indicated a higher overall seroprevalence in this target population. Our results show that the protective measles IgG values were found in 91.7% of the tested population. This level is higher than that reported (88%) in a previous meta-analysis that investigated the susceptibility to measles in Italian HCWs [22], reporting a total susceptibility of 12%, with several differences across the different regions. The reported susceptibility is, moreover, higher than the overall rate of the susceptibility reported in European HCWs (6%) but similar to that reported in Spain (11%) [23,24]. These data do not report the percentage of unvaccinated people, but another serosurvey conducted in an Italian cohort of 2000 adults has shown that 15.3% of fully vaccinated individuals were still susceptible to measles [25]. Moreover, the overall measles seroprevalence among HCWs in South Korea (92%) is similar to the present results. South Korea’s experience with a measles epidemic in 2019, where more than two-thirds of cases were caused by in-hospital transmission, despite the WHO declaring this disease to have been eliminated since 2014, suggests that attention needs to be maintained regarding vaccine-preventable diseases [26,27].

Varicella IgG was present in 83.9% of our sample, similar to results reported in a national study which found an overall seroprevalence of 84%. This is significant as in Italy varicella has been and remains the most important vaccine-preventable infectious disease. Moreover, for many years, despite the accessibility to vaccines, varicella vaccination has not been recommended and effected nationwide [28]. The present data demonstrate a higher seroprevalence rate than that reported in a similar population in Italy, showing almost 34% of the participants did not have circulating anti-varicella IgG [29]. Moreover, our sample has a lower seroprevalence than those reported in a similar population in China, where the overall seroprevalence rate of varicella among HCWs was 88.4% [30], and in Korea, where the seroprevalence of varicella was over 90% [31]. Our results confirm those of other similar studies regarding lower seroprevalence in younger people [30]. Indeed, it has been demonstrated that varicella seroprevalence decreases over time among the vaccinated population, but a booster dose of vaccine results in a significantly higher titer [29,30,31,32]. These data encourage varicella seroprevalence surveillance and vaccination among HCWs and it are consistent with results reported for the hepatitis B vaccination, confirming that adolescents and adults who get a booster dose were more likely to be seroprotected, and that the longer the time elapsed since vaccination, the lower the probability of being seropositive [33].

Rubella IgG were present in 82% of the analyzed population. This rate is lower than that reported in a recent meta-analysis which estimated the coverage for rubella among HCWs in Italy at 91% [34]. Our findings reported a higher seroprevalence for rubella in males. This result is different to that reported in other studies with a similar population, which have found that females have a higher level of immunization than males, probably as a consequence of the historical immunization campaign targeted at preventing the risk of congenital rubella [34,35]. Two articles described a higher proportion of serosusceptibily in female nurses than male nurses [36,37]. In this instance as well, the proportion of serosusceptibility does not indicate unvaccinated people; indeed, it has been reported that [38] 23.6% of individuals who had received two boosters of vaccine continued to have no antibodies. The proportions of HCWs totally vaccinated among those serosusceptible ranged from 13% to 25% [37,39]. Regarding this point, Coppeta et al. [36] suggest achieving a higher uptake among female nurses through MMR vaccination campaigns. Coppeta et al. also note that male nurses, who were more often not vaccinated, were more likely to have a longer-lasting natural immunity, which gives them, unexpectedly, higher seroprotection.

The percentage of seroprevalence for mumps is 76.2%. This rate is lower than that reported in HCWs in South Korea (87%) [40] and Denmark (86.5%) [15], but higher than that reported for new nurses in Korea 60.2% [41] and Japan (54.1%) [42]. Numerous serological studies have explained that mumps antibody levels reduce with time since vaccination [43,44]. Moreover, it has been demonstrated that the mumps vaccine component of MMR provides a lower immune response than measles and rubella constituents [45]. Studies conducted in Northern Europe have documented the presence of antibodies in children derived from the Jeryl Lynn mumps strain, reporting only 73% of people have had seroconversion after a single dose of vaccination [46]. Ferrari et al. [14] have evaluated the presence of a protective antibody level for mumps in a population of medical students vaccinated during childhood or adolescence in Italy, observing that a significant proportion of the students lacked serological protection from mumps.

Knowing who is not seropositive for infectious diseases for which a vaccine is available is important because vaccination, and the consequent immunization of this population, can avoid the initiation of infectious hospital clusters. Although there are no precise indications in the literature about the treatment of people susceptible to infection, the full vaccination cycle [47,48] or post-exposure prophylaxis with immunoglobulin seems to be the most widely described in the literature [25].

Other studies reported that vaccinating susceptible HCWs may give an opportunity to reduce the risk of hospital clusters [49,50]. Indeed, [38] it has been suggested to monitor individuals 10–15 years after vaccination, in order to revaccinate those who were seronegative. The recent experience gained during the COVID-19 pandemic, in which Italy was among the most affected nations, has shown how the monitoring of seroprevalence among HCWs is essential to reduce the spread of transmissible infections. Moreover, it provided evidence of the usefulness of repeated seroprevalence surveys, especially among HCWs, in order to protect patients [51,52].

### Limitations

Several limitations should be acknowledged in the analyses of the results of this survey. First, a convenience sample was recruited, and external validity is therefore not guaranteed, since the impact of the willingness to participate and the representativeness of the recruited population on seroprevalence values are difficult to determine. Second, the test sample is small and not homogeneous by sex. Third, we do not know the immune history of the subjects and whether the immunized subjects underwent vaccination, at how many doses, and when. Therefore, we cannot determine whether the presence of antibodies is due to active immunization or disease. Likewise, we cannot compare the level of antibodies present after vaccination or after disease and we are not able to determine the percentage of unresponsiveness for vaccinations carried out. Moreover, this study investigates a population of trainee obstetricians in a single hospital in Southern Italy, so the results are not generalizable to other trainee HCWs (nurses, physicians, etc.), to the entire population of HCWs in our area, or to the whole population.

Furthermore, serologic tests are limited by mistakes with false-positive or false-negative errors, with the false positives being of more interest in populations with an estimated low seroprevalence. However, the involved tests had a very high stated sensitivity and specificity, and we may be confident that the false positives and negatives were insufficient to have a consistent impact on the closing estimation.

## 5. Conclusions

The data obtained in the population tested to show that a percentage of trainee obstetrics do not have vaccination coverage against at least one of the vaccine-preventable childhood diseases, especially for mumps. Therefore, it is proposed to implement both a seroprevalence study in hospital wards at greater risk by extending surveillance to other professionals in contact with frail patients, and vaccination campaigns to increase immunity levels. The screening of future and employed HCWs is essential to prevent hospital clusters; the promotion of an adequate immunization program should be a priority of Occupational Medicine Services. It becomes fundamental to verify the serological state of HCWs and implement the appropriate measures of prophylaxis in seronegative cases in order to prevent MMR hospital infection. The immunity gaps found primarily in future obstetricians indicate a need for a screening program to identify those who would benefit from vaccination and public health strategy for this group. Obstetricians and obstetrical assistants should be vaccinated against measles, varicella, mumps, and rubella and should be subject to antibody screening not only for their safety but for that of pregnant women, fetuses, and newborns.

The updating of epidemiologic data is decisive for an overall assessment of the impact of vaccination against measles, varicella, mumps, and rubella, both with regards to the reporting of cases, complications, hospitalizations, and deaths due to disease impact, as well as seroprevalence changes due to program implementation of vaccination strategies.

## Figures and Tables

**Table 1 vaccines-12-00335-t001:** Socio-demographic and laboratory characteristics in the trainee obstetric group.

Characteristics	Total	(n.517)
Socio-Demographic Characteristics	N	%
**Gender**		
Male	67	13
Female	450	87
**Age groups, (years)**	26.3 ± 5.7 (range 19–53) *	
≤23	209	40.4
24–29	172	33.3
30–36	114	22.1
≥37	22	4.3
**Laboratory characteristics**		
**Date of collection**		
2017	48	9.3
2018	37	7.2
2019	137	26.5
2020	3	0.6
2021	166	32.1
2022	126	24.4
**Measles**		
Sieronegative	27	5.2
Sieropositive	474	91.7
Equivocal	16	3.1
**Varicella**		
Sieronegative	63	12.2
Sieropositive	434	83.9
Equivocal	20	3.9
**Rubella**		
Sieronegative	37	7.2
Sieropositive	424	82
Equivocal	56	10.8
**Mumps**		
Sieronegative	69	13.3
Sieropositive	394	76.2
Equivocal	54	10.4

* Mean ± Standard Deviation (Range).

**Table 2 vaccines-12-00335-t002:** Immunity against measles, varicella, rubella, and mumps in the trainee obstetrics.

Measles			
	Susceptible n (%)	Immune n (%)	*p*
**Age (in years)**	26.3 ± 0.7 *	26.3 ± 0.3 *	0.968
**Age groups**			
≤23	19 (9.1)	190 (90.9)	0.208
24–29	11 (6.4)	161 (93.6)	
30–36	12 (10.5)	102 (89.5)	
≥37	1 (4.5)	21 (95.5)	
**Gender**			
Male	5 (7.5)	62 (92.5)	0.543
Female	38 (8.4)	412 (91.6)	
**Total**	43 (8.3)	474 (91.7)	
**Varicella**			
**Age (in years)**	24.7 ± 0.2 *	26.6 ± 0.03 *	0.005
**Age groups**			
23	48 (23)	161 (77)	0.011
24–29	24 (14)	148 (86)	
30–36	7 (6.1)	107 (93.9)	
≥37	4 (18.2)	18 (81.8)	
**Gender**			
Male	6 (9)	61 (91)	0.217
Female	77 (17)	373 (83)	
**Total**	83 (16.1)	434 (83.9)	
**Rubella**			
**Age (in years)**	24.8 ± 0.4 *	26.6 ± 0.3 *	0.004
**Age groups**			
≤23	45 (21.5)	164 (78.5)	0.006
24–29	34 (19.7)	138 (80.3)	
30–36	11 (9.6)	103 (90.4)	
≥37	3 (18)	19 (82)	
**Gender**			
Male	8 (11.9)	59 (88.1)	0.200
Female	85 (18.9)	365 (81.1)	
**Total**	93 (18)	424 (82)	
**Mumps**			
**Age (in years)**	25.3 ± 0.5 *	26.6 ± 0.3 *	0.031
**Age groups**			
≤23	61 (29.2)	148 (70.8)	0.089
24–29	40 (23.3)	132 (76.7)	
30–36	17 (14.9)	97 (85.1)	
≥37	5 (22.7)	17 (77.3)	
**Gender**			
Male	14 (20.9)	53 (79.1)	0.427
Female	109 (24.2)	341 (75.8)	
**Total**	123 (23.8)	394 (76.2)	

* Mean ± Standard deviation.

## Data Availability

The data presented in this study are available on request from the corresponding author.

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
