# Peer review of "Seroprevalence for Measles, Varicella, Mumps and Rubella in the Trainee Obstetric Population: A Survey in Southern Italy"

_vaccines, 2024, doi:10.3390/vaccines12030335_

Round 1

Reviewer 1 Report

Comments and Suggestions for Authors

1. In the last sentence of line 69, "Each year born about 105.000 children with CRS.", it is necessary to specify the range of people population involved in this data, is it worldwide? All over Europe? Just in Italy? Or is it in West of Romania?

2. Line 124, the abbreviation for "enzyme-linked immunosorbent assay" is common expressed as ELISA, what is the author's reason for using EIA?

3. Vaccination of obstetricians and obstetrical caregivers against measles, varicella, mumps and rubella is important not only for their own safety, but also for the biosafety of pregnant women, fetuses and newborn children. Based on the statistical analysis of the study, the authors are suggested to give such an opinion or conclusion in this paper: it is necessary for obstetricians and obstetrical caregivers to enforce measles, varicella, mumps and rubella vaccine vaccination and regular antibody positive rate detection.

4. According to the description by the authors in the introduction of this study, are pregnant women advised to receive measles, varicella, mumps and rubella vaccine vaccination at the right time to protect the fetuses and newborn children?

Author Response

Dear Reviewer,

Thank you for having reviewed the manuscript: “Seroprevalence for measles, varicella, mumps and rubella in the trainee obstetric population: a survey in Southern Italy”. We have fully addressed your concerns; below and in the new document we have indicated all the revisions made.

1. In the last sentence of line 69, "Each year born about 105.000 children with CRS.", it is necessary to specify the range of people population involved in this data, is it worldwide? All over Europe? Just in Italy? Or is it in West of Romania?As suggested, we have specified the range of population involved in this data. Moreover, we have reported a more recent data, reference (7) has been replaced with reference Gorun et al., 2021.

2. Line 124, the abbreviation for "enzyme-linked immunosorbent assay" is common expressed as ELISA, what is the author's reason for using EIA?
As suggested, we replaced the abbreviation " EIA " with "ELISA".

3. Vaccination of obstetricians and obstetrical caregivers against measles, varicella, mumps and rubella is important not only for their own safety, but also for the biosafety of pregnant women, fetuses and newborn children. Based on the statistical analysis of the study, the authors are suggested to give such an opinion or conclusion in this paper: it is necessary for obstetricians and obstetrical caregivers to enforce measles, varicella, mumps and rubella vaccine vaccination and regular antibody positive rate detection.
As suggested, in “conclusion section” we stressed the importance of implementing adherence to vaccinations by midwives and obstetrical caregivers.

4. According to the description by the authors in the introduction of this study, are pregnant women advised to receive measles, varicella, mumps and rubella vaccine vaccination at the right time to protect the fetuses and newborn children?
As suggested, in the introduction (line 40-41) we added a sentence to clarify that women of childbearing age should be vaccinated against chickenpox, measles and rubella at least 1 month, 3 months and 6 months before conception.

My colleagues and I are most grateful for the extremely positive tone of the reviewer’s comments.

Yours sincerely.

Reviewer 2 Report

Comments and Suggestions for Authors

I have read with interest this manuscript which reports on the sero-prevalence of measles, rubella and varicella among Italian Health Care Workers.

The topic  is interesting given the increase in measles cases in Europe. However, the manuscript presents several limitations which need to be addressed by the authors.

1. Introduction.

The authors should provide the reader with a brief introduction to the results of sero-prevalence studies among health care workers (measles, rubella, pertussis). 

2. Methods.

I am not clear about equivocal test results especially for rubella. Did  an equivocal result prompt collection of a new sample? Have the equivocal results been included in the statistical analysis? I believe that they should be excluded for further analysis. In statistical analysis the authors used age group categories. I would prefer to treat age as a continuous variable and compare the mean between positives and negatives.  

3. Results.

Table 3 is confusing.

4. Discussion.

The authors should critically discussed their results in relation to other previously published studies among health care workers. The have devoted disproportionate space discussing results from general population samples.

The response rate of the survey should be reported in both abstract and results. 

Comments on the Quality of English Language

Minor editing required.

Author Response

Dear Reviewer,

Thank you for having reviewed the manuscript: “Seroprevalence for measles, varicella, mumps and rubella in the trainee obstetric population: a survey in Southern Italy”. We have fully addressed your concerns; below and in the new document we have indicated all the revisions made.

I have read with interest this manuscript which reports on the sero-prevalence of measles, rubella and varicella among Italian Health Care Workers. The topic  is interesting given the increase in measles cases in Europe. However, the manuscript presents several limitations which need to be addressed by the authors.

1. Introduction.

The authors should provide the reader with a brief introduction to the results of sero-prevalence studies among health care workers (measles, rubella, pertussis).
As suggested, in the Introduction section, we have reported a brief introduction regarding the sero-susceptibility among HCWs (line 107-108).

2. Methods.

I am not clear about equivocal test results especially for rubella. Did  an equivocal result prompt collection of a new sample? Have the equivocal results been included in the statistical analysis? I believe that they should be excluded for further analysis. In statistical analysis the authors used age group categories. I would prefer to treat age as a continuous variable and compare the mean between positives and negatives.
In response to this point, we agree with the reviewer, indeed the equivocal results had been included in the statistical analysis as “Susceptible”. Generally in cases of doubtful results in the patient’s report is recommended to repeat the investigation with a new blood sample and considered susceptible. As suggested, we provided to analyze age as continuous variable in Table 2.

3. Results.

Table 3 is confusing.

In response to this point, we have deleted the Table 3 and summarized results in Table 2 to avoid misunderstanding.   

4. Discussion.

The authors should critically discussed their results in relation to other previously published studies among health care workers. The have devoted disproportionate space discussing results from general population samples.
As suggested, we have reduced discussion of results from general population samples and added discussion among HCWs.

5. The response rate of the survey should be reported in both abstract and results.
As suggested, we have reported the response rate in both abstract and results.

My colleagues and I are most grateful for the extremely positive tone of the reviewer’s comments.

Yours sincerely.

Round 2

Reviewer 2 Report

Comments and Suggestions for Authors

The authors stated in their rebuttal letter that an equivocal test result resulted in the assumption that the subject was "susceptible". This important statement should be included in the methods section of the manuscript.

Page 7; line 179: Please delete the word "significantly". 

Comments on the Quality of English Language

Minor editing is required.

Author Response

Thank you for having reviewed the manuscript: “Seroprevalence for measles, varicella, mumps and rubella in the trainee obstetric population: a survey in Southern Italy”. We have fully addressed your concerns; below and in the new document we have indicated all the revisions made.

The authors stated in their rebuttal letter that an equivocal test result resulted in the assumption that the subject was "susceptible". This important statement should be included in the methods section of the manuscript.

As suggested, we have included this statement in the methods section.

Page 7; line 179: Please delete the word "significantly". 

As suggested, the word “significantly” has been deleted.

My colleagues and I are most grateful for the extremely positive tone of the reviewer’s comments.

Yours sincerely.